



# Estuarine flocculation dynamics of organic carbon and metals from boreal acid sulphate soils

Joonas J. Virtasalo[1], Peter Österholm[2], Eero Asmala[3]

[1]Marine Geology, Geological Survey of Finland (GTK), Espoo, 02150, Finland
[2]Department of Geology and Mineralogy, Åbo Akademi University, Turku, 20500, Finland
[3]Environmental Geochemistry, Geological Survey of Finland (GTK), Espoo, 02150, Finland

*Correspondence to*: Joonas J. Virtasalo (joonas.virtasalo@gtk.fi), Eero Asmala (eero.asmala@gtk.fi)

**Abstract.** Flocculation of riverine dissolved organic matter to the particulate form in estuaries is an important mechanism for capturing dissolved metals to newly formed organic particles, regulating the metal transport to the sea. The process is

particularly relevant for rivers draining boreal acid sulphate soils of western Finland, which are known to deliver large amounts of trace metals with detrimental environmental consequences to the recipient estuaries in the eastern Gulf of Bothnia, northern Baltic Sea. This is the first study to investigate dissolved metal (Al, Fe, Mn, Co and Cu) association with flocculating organic particles in the laboratory, by mixing of acidic metal-rich water from acid sulphate soil-impacted rivers and particle-free artificial seawater. Water samples were collected in April 2021 from the Laihianjoki and Sulvanjoki rivers in western Finland.

Experiments with an *in situ* laser diffraction-based particle size distribution sensor and a multiparameter water quality sonde were run to continuously monitor the development of suspended particle pool over the salinity gradient from 0 to 6, corresponding the salinity range observed in these estuarine systems. Flocculator experiments with discrete salinity treatments were carried out to investigate metal behaviour with the collection of flocculated material on glassfibre filters. Filtrate was analysed for coloured dissolved organic matter absorbance and fluorescence for the characterization of potential changes in

the organic matter pool during the flocculation process. Retentate on the filter was subjected to persulfate digestion of organic particles and metal oxyhydroxides (pH <2.3), and the digestion supernatants were analysed for metal concentrations. The laboratory experiments showed strong transfer of Al and Fe already at salinity 0–2 to newly formed organic-dominated flocs that were generally larger than 80 µm. Very strong coupling between the decrease in humic-like fluorescence and the increase in organic-bound Al demonstrated that Al transfer to the flocs was stronger than that of Fe. The flocs in the suspended particle

pool were complemented by a smaller population of Al and Fe oxyhydroxide-dominated flocculi (median size 11 µm) after pH exceeded ca. 5.5. Cobalt and Mn transfer to the particle pool was weak, although some transfer to Mn oxyhydroxides as well as Co association with the flocs took place. Up to 50 % of Cu was found to be bound to humic substances in the flocs in the river waters and this proportion did not significantly change during mixing with seawater. The findings of this study demonstrate that salinity and pH are important independent but connected controls of the flocculation behavior of dissolved

metals from boreal acid sulphate soils and the seaward transport and environmental consequences of the metals in the marine environment.



# 1 Introduction

Among the most important processes affecting the global biogeochemical cycles of carbon, nutrients and trace metals is the physicochemical transformation of terrestrial dissolved and colloidal material into suspended particles at the freshwater and

seawater mixing zone in estuaries (Boyle et al., 1974; Bianchi, 2007; Canuel and Hardison, 2016). The influence of seawater cations induces the flocculation of riverine dissolved organic matter (DOM) to the particulate form, which provides a mechanism for the capture of dissolved metals that have a tendency to be adsorbed to particle surfaces (Sholkovitz, 1976; 1978; Asmala et al., 2022). In addition to salinity, also pH typically increases towards seawater, and provides another key variable that controls metal sorption to particle surfaces (Mosley and Liss, 2020). The role of pH is particularly relevant in the

context of acid sulphate (AS) soils that can generate extremely low pH conditions and high trace metal concentrations in surface waters (Dent and Pons, 1995). Flocculation of dissolved riverine material into sinking particles is a key component of the coastal filter, which effectively removes terrestrial material from the freshwater inputs in the coastal salinity gradient (Asmala et al., 2017).

AS soils occupy globally ca. 50 million ha, including areas in Australia, South and Southeast Asia, Africa, Central and South America, and Northern and Western Europe (Andriesse and van Mensvoort, 2006). The largest areas of AS soils in Europe are found in Finland, where they occupy an area in the order of 1 million ha (Jaakko Auri, personal communication, 2022). The Finnish AS soils are predominantly sulphide-bearing organic-rich muds, which were originally deposited in the Baltic Sea during its brackish-water phase, which began ca. 7000 years ago in the Gulf of Bothnia (Virtasalo et al., 2007; Häusler et al.,

2017), but these coastal areas have since emerged above sea level due to rapid glacio-isostatic land uplift (Kakkuri, 2012). Artificial drainage and reclamation of these lands for farming purposes has caused a significant lowering of the groundwater level, which has enabled rapid oxidation of the sulphide minerals, producing $H_2SO_4$ and resulting in AS soils with a pH < 4 (Yli-Halla et al., 1999; Sohlenius and Öborn, 2004; Boman et al., 2010). Under these extremely acidic conditions, large quantities of metals are released to the porewater due to the oxidative dissolution of metal sulphides and the weathering of

silicate minerals. Particularly during high water flow conditions in spring and autumn, acidic porewater rich in metals (typically Al, Cd, Co, Cu, Mn, Ni, and Zn) is flushed to recipient streams (Österholm and Åström, 2008), with detrimental ecological consequences for aquatic life (Hudd and Kjellman, 2002; Fältmarsch et al., 2008; Wallin et al., 2015; Sutela and Vehanen, 2017).

Previous field studies and geochemical modelling results in the west coast of Finland have concluded that when acidic metal-rich river waters from the boreal AS soil landscape are discharged to estuaries with higher pH and salinity, the metals are complexed with organic matter or co-precipitated with Al, Fe, and Mn oxyhydroxides and consequently deposited in sediments (Nordmyr et al., 2008a, b; Nystrand et al., 2016). Specifically, it was concluded that Al, Fe and Cu are deposited with organic matter close to the Vöyrinjoki river mouth, whereas Cd, Co, Ni, and Zn are preferentially transported farther out at the sea,

where they co-precipitate and are deposited with Mn oxyhydroxides (Nordmyr et al., 2008a, b; Nystrand et al., 2016). However, a recent study of seafloor sediments shows that Cu as well as Cd, Co, Ni, and Zn all are enriched at least 26 km away from the mouths of the nearby Laihianjoki and Sulvanjoki rivers, whereas the Mn enrichment extends less than 14 km from the river mouths (Virtasalo et al., 2020). This discrepancy between the observed seafloor metal distribution and the previously concluded seaward transport behaviour of the metals calls for further investigation of the flocculation process in boreal humic-

rich rivers that are strongly impacted by acidity and metal release from AS soils.

This study investigated organic matter flocculation and aggregation, and trace metal association with the newly formed organic particles. To address this phenomenon, we carried out laboratory experiments, mixing natural acidic river water with particle-free artificial seawater. River water was collected from the Laihianjoki and Sulvanjoki rivers that are among the most AS-soil-

impacted rivers in Finland and Europe (Roos and Åström, 2005). Large-volume bucket experiments with high-frequency *in situ* instruments were continuously run to study the development of suspended particle size distribution and organic matter characteristics, and to identify possible threshold salinities for further experimentation. Small-volume flocculator experiments with discrete salinity treatments were carried out to investigate metal behaviour in the flocculation with subsequent laboratory analyses of the flocculated material. The focus was on Al, Fe and Cu, which, according to previous studies in the area, to a

large extent form complexes with flocculating organic matter and precipitate as Al and Fe oxyhydroxide minerals in the early phases of freshwater-seawater mixing, as well as on Co and Mn that are relatively persistent in solution (Nordmyr et al., 2008a, b; Nystrand et al., 2016). The studied elements Al, Mn, Co, and Cu are known to be extensively leached from AS soils and much less so from the local industry and other human activities in the area (e.g., Åström and Björklund, 1997; Åström and Corin, 2000; Österholm and Åström, 2002; Sundström et al., 2002). Similarly, background concentrations of Fe are typically

very high in coastal areas of Finland (e.g., Lahermo et al., 1996).

The objectives of this investigation were to (1) assess the quantity of dissolved metal elements removed from acidic and metal-rich river water by association with flocculating organic particles and (2) quantify changes in the size distribution of suspended particles due to increasing salinity and pH. We hypothesize that (1) with increasing salinity and pH, the proportion of dissolved

metals that are associated with organic particles increases, and (2) the size of suspended particles increases throughout the experimental salinity gradient as dissolved substances and smaller particles aggregate into larger particles.

## 2 Study area

The Laihianjoki and Sulvanjoki rivers are located in the central western Finland, where they drain into the shallow Sundominlahti bay near the town of Vaasa in the Kvarken archipelago of the Gulf of Bothnia (Fig. 1). Laihianjoki is 60 km

long and has a catchment area of 506 km$^2$ with an overall flat topography that is dominated by forests and peatlands (61 %) and agricultural land (27 %; Table 1; Korhonen and Haavalammi, 2012). Sulvanjoki is only 33 km long and drains a catchment





area of 144 km² with a clearly larger share (36 %) occupied by agricultural land. In both catchments, most of the agricultural land consist of well-drained fine grained AS soils that leach large quantities of acidity and metals. Due to the significantly higher proportion of AS soils in the Sulvanjoki catchment area, its acidity and metal concentrations are expected to be higher than in Laihianjoki.

Crystalline bedrock in the study area is dominated by Paleoproterozoic granitoids and gneisses that are overlain by an average 5 m thick till layer and occasional till ridges (De Geer moraines, drumlins) and glaciofluvial landforms (Breilin et al., 2005). Following the north-westward retreat of the Fennoscandian continental ice sheet ca. 10 400 calendar years BP, the study area was submerged by a ca. 250 m deep lake (Sauramo, 1929; Saarnisto and Saarinen, 2001; Stroeven et al., 2016). A glacial varved to weakly-layered mud blanket was deposited on the till and glaciofluvial deposits during the successive glaciolacustrine and post-glacial lake phases. The lake sediments are sharply overlain by an organic-rich mud layer that has deposited after the onset of the modern brackish-water sea phase ca. 7000 cal. yr. BP (Virtasalo et al., 2007; Häusler et al., 2017). The area has since emerged above sea level as a result of the glacio-isostatic rebound that continues today at a notable rate of 8 mm yr⁻¹ (Kakkuri, 2012). During the emergence, mud was removed by coastal erosion from the local topographic highs and redeposited in local depressions. Today, agricultural fields in the area are largely developed on the lowland areas with brackish-water sulphide-bearing organic-rich muds that are active or potential AS soils (Yli-Halla et al., 1999; Boman et al., 2010).

The study area belongs to the continental subarctic climate zone with severe dry winters and warm summers. The mean annual air temperature is 4.2 °C, with the mean minimum temperature of 2.1 °C and the mean maximum temperature of 6.6 °C during the period 1981–2010 (Pirinen et al., 2012). The mean annual precipitation is 497 mm. The annual mean sea surface salinity in the Kvarken archipelago ranges between 3.5 and 4, and the annual mean sea surface temperature is between 3.5 and 7 °C. The low salinity results from the high riverine runoff from the large catchment area of the Gulf of Bothnia, and from the long distance to the narrow connection to the North Sea through the Danish straits (Kuosa et al., 2017). The sea is essentially non-tidal, but irregular water level fluctuations of as much as ±1.5 m take place due to variations in wind and atmospheric pressure (Wolski et al., 2014). Stratification of the shallow archipelago waters is governed by a thermocline that develops each summer. Seasonal ice-cover in the inland and coastal waters is typical during winter months.

## 3 Materials and methods

### 3.1 Field sampling

Water samples were collected from the Laihianjoki and Sulvanjoki rivers in April 2021 using a bucket and poured into 30 L containers (Table 1). Springtime, several weeks after the flow peak induced by snow smelt, was chosen for the sampling, because then the variation in water quality in AS soil-impacted rivers is the smallest and most representative samples can be



obtained (Österholm and Åström, 2008). The river waters were immediately measured for pH, temperature, electrical

conductivity and salinity by a WTW Multiline P4 meter (WTW GmbH, Weilheim, Germany). The water samples were transported under cool conditions to the Tvärminne field station of the University of Helsinki, where the laboratory experiments commenced within 24 h.

## 3.2 Experimental set-ups

A large-volume "bucket" experiment was carried out to determine the dynamics of suspended particle size distribution and

associated nutrients during the gradual transition from fresh to brackish water conditions following the experimental protocol described by Asmala et al. (2022). In short, a salt solution was prepared by diluting artificial sea salt (Aquarium Systems Instant Ocean) to ultrapure water and filtered through pre-combusted glass fiber filter (nominal poresize 0.7 µm) to remove any possibly remaining particles. The final salinity of the solution was 66.3 g kg$^{-1}$. The experiment was conducted on Laihianjoki and Sulvanjoki river waters, screened with a 200 µm nylon mesh for removing debris. River water was poured

into an acid-washed basin (bucket) with a mounted spinning device rotating at a fixed speed of ca. 100 RPM to ensure proper mixing throughout the experiment. Changes in water chemistry and suspended particle size distribution were recorded continuously with an EXO2 multiparameter water quality sonde (YSI, Yellow Springs, OH, USA) and a LISST-100X type B laser-diffraction based particle size distribution analyzer (Sequoia Scientific, Bellevue, WA, USA). The EXO2 multiparameter sonde was equipped with six electronic and optical sensors, measuring conductivity (mS cm$^{-1}$), temperature (°C), dissolved

oxygen (mg L$^{-1}$), pH, fluorescent dissolved organic matter (FDOM in QSU) and turbidity (FNU). The LISST particle size sensor measures suspended particle (aggregate) size distribution and concentration *in situ* over 32 size classes (bins) in the range from 1.25 to 250 µm (Agrawal and Pottsmith, 2000). Both instruments were calibrated using manufacturers' guidelines, and the measuring interval for both instruments was 1 Hz. The artificial seawater was gradually added using a peristaltic pump and acid-washed tubing to the experimental basin containing the river water, in order to simulate an increasing estuarine

salinity gradient from 0.0 to 6.0 g kg$^{-1}$. All measured concentration values were corrected for the dilution by the particle-free artificial seawater in subsequent analyses.

After the bucket experiment, LISST data were processed using the random particle shape model in the instrument data processing software. Weighted mean particle size was calculated using the median size of each LISST size class. The values

for salinity, temperature, pH, FDOM and turbidity were determined from the factory-calibrated sensor readings by the integral algorithms of the EXO2 multiparameter sonde software.

A set of small-volume jar experiments was carried out to collect particle samples from standardized and replicable conditions using a Velp Scientifica FC6S 6-unit flocculator apparatus (VELP Scientifica, Usmate, Italy). Same artificial seawater was

used as in the bucket experiment with *in situ* instruments. Sample water from study rivers was mixed with ultrapure water and artificial seawater to create a salinity gradient from 0 to 6 g kg$^{-1}$ to a final volume of 1000 mL. The flocculator test duration





was 1 hour at a fixed paddle speed of 60 RPM. At the end of the test cycle, suspended material was collected on triplicate precombusted glassfibre filters with a nominal pore size of 0.7 µm. Filtrate was collected for CDOM analysis.

### 3.3 Laboratory analyses

Colored DOM (CDOM) absorption was measured using a Shimadzu 2401PC spectrophotometer (Shimadzu, Kyoto, Japan) with 5 cm quartz cuvette over the spectral range from 200 to 800 nm with 1 nm resolution. Ultrapure water was used as the blank for all samples. Excitation-emission matrices (EEMs) of fluorescent DOM (FDOM) were measured with a Varian Cary Eclipse fluorometer (Agilent Technologies, Santa Clara, CA, USA). Processing of the EEMs was done using the *eemR* package for R software (Massicotte, 2016). A blank sample of ultrapure water was subtracted from the EEMs, and the Rayleigh and

Raman scattering bands were removed from the spectra after calibration. EEMs were calibrated by normalizing to the area under the Raman water scatter peak (excitation wavelength of 350 nm) of an ultrapure water sample run on the same session as the samples, and were corrected for inner filter effects with absorbance spectra (Murphy et al., 2010). For assessing the characteristics and the quality of the DOM pool, fluorescence peaks (Coble, 1996) were calculated from the EEMs (peak C – humic-like; peak T – protein-like).


Triplicate samples of the Laihianjoki and Sulvanjoki river waters and the artificial seawater were acidified by $HNO_3$ and analyzed for Al and Fe concentrations by a Thermo Scientific iCAP 7600 Duo inductively coupled plasma optical emission spectrometer (ICP-OES), and for Co, Cu and Mn concentrations by a Thermo Scientific iCAP Q inductively coupled plasma mass spectrometer (ICP-MS; Thermo Fisher Scientific Inc., Waltham, MA, USA). The detection limits for Al, Co, Cu, Fe and

Mn were 200, 0.05, 0.1, 30 and 0.02 µg $L^{-1}$, respectively.

For retentates of filtered samples, three separate analyses were carried out: particulate organic carbon (POC), particulate organic-bound metals and total suspended matter (TSM). POC analysis was performed using Europa Scientific ANCA-MS 20-20 15N/13C mass spectrometer (Sercon, Crewe, UK) from filter samples. Total digestion of organic particles on triplicate

filter samples was performed by persulfate oxidation according to Pujo-Pay and Raimbault (1994). Persulfate digestion results in a pH <2.3, which is thought to effectively dissolve metal oxyhydroxides (Schwertmann, 1991). Al, Co, Cu, Fe and Mn concentrations in the filter digestion supernatants were analyzed by ICP-MS, controlled with blank reagent samples. The detection limits for Al, Co, Cu, Fe and Mn were 20, 0.2, 10, 100 and 1 µg $L^{-1}$, respectively. Values below detection were rounded to half the detection limits. The final element concentrations were then calculated by subtracting the concentrations

detected on the equipment blank samples.

TSM was determined by filtering a known volume of sample water through pre-weighed GF/F filter, and subsequently drying the filters overnight at 60 °C and then weighing again. The apparent particle density (Thomas et al., 2017) was calculated by dividing TSM (mass concentration; in mg $L^{-1}$) with the particle total volume concentrations measured by LISST (volumetric



concentration; in µL L$^{-1}$) from matching salinities in bucket and flocculator experiments. It should be noted, however, that the apparent particle density represents only particles measurable by the LISST instrument, i.e. particles within the hydrodynamic radius between 1 and 250 µm. As the proportion of particles in the sample that are larger than our upper detection limit is non-trivial, it can be expected that our density results are biased as the largest particles are not included in the analysis.

## 4 Results

### 4.1 River water initial composition

Samples from Sulvanjoki had higher total concentrations of Al, Co and Mn, similar concentrations of Cu and lower concentrations of Fe compared to Laihianjoki (Table 2).

### 4.2 Large-volume bucket experiment

### 4.2.1 Transformation of dissolved substances into particulate form

Initial characteristics of the particulate and dissolved organic matter pools were different between the study rivers (Fig. 2). In Laihianjoki, the mean particle size was 84 µm and the total volume concentration of 1–250 µm particles was 201 µL L$^{-1}$, whereas in Sulvanjoki the values were higher; 130 µm and 266 µL L$^{-1}$, respectively. On the other hand, initial turbidity was higher in Laihianjoki (25 FNU) than in Sulvanjoki (10 FNU). The initial DOM concentration, as indicated by humic-like fluorescence, was 90 QSU in Laihianjoki and 126 QSU in Sulvanjoki.


Turbidity and total particle volume concentration increased throughout the experimental salinity gradient in both rivers (Fig. 2). In Laihianjoki, turbidity increased to 37 FNU and particle volume concentration to 880 µL L$^{-1}$ at salinity 5.0, and in Sulvanjoki to 34 FNU and 1300 µL L$^{-1}$, respectively. It should be noted that the rate of increase was considerably lower above salinity ~2 in Laihianjoki. Conversely, humic-like fluorescence decreased in both rivers; to 55 QSU in Laihianjoki and to 63

QSU in Sulvanjoki. Mean particle size in Laihianjoki showed a dynamic response to seawater additions, displaying peak value of 115 µm at salinity 1, followed by a steady decline to a value of 100 µm at salinity 5 (Fig. 2). In contrast, the mean particle size sustained at a high level in Sulvanjoki across the experimental salinity gradient, showing only a small decrease from 130 µm to 120 µm in the salinity range 3–5. During the bucket experiment, the seawater addition caused an increase in pH; from the initial 5.2 to 6.9 in Laihianjoki and from 5.2 to 6.3 in Sulvanjoki. pH 5.5 was reached at salinity 0.5 in Laihianjoki, while

it was reached later in Sulvanjoki at salinity 1.4.

### 4.2.2 Changes in suspended particle size distribution

Development of particle volume concentrations in the 32 size classes during the bucket experiment was measured by the LISST particle size distribution analyzer (Fig. A1 in the Appendix). Two main types of behavior were identified by visual inspection

among the size classes: the finer classes with the particle median sizes of 9.35 and 11.0 µm (bins 14 and 15) showed low initial
values until the concentrations started to increase strongly at salinity 1 (pH 5.75) in Laihianjoki and at salinity 2 (pH 5.59) in
Sulvanjoki, whereas in the coarsest size classes with particle median sizes of 156.0 and 184.1 µm (bins 31 and 32), the
concentrations began to increase strongly from the start of the experiment at salinity 0 (pH 5.30 in Laihianjoki and 5.33 in
Sulvanjoki). The strong initial increase in the coarsest particle concentrations levelled off and turned to decrease at salinity 2
(pH 6.16) in the Laihianjoki river. The finest size classes (1.0–8.6 µm; bins 1–13) each displayed very low volume
concentrations of less than 1 % of the coarsest class; these finest classes were considered insignificant for the studied
flocculation process and were not considered further.

The size classes 11.0 and 184.1 µm (bins 15 and 32 with the ranges of 10.2–12.0 and 169–200 µm, respectively) were selected
for further investigation of the development of suspended particle size distribution during the bucket experiment (Fig. 3). The
size class concentration ratio of 184/11 µm shows a dominant initial increase of the coarsest particles in the flocculation in the
salinity range from 0 to 1. Above salinity 1, however, the proportion of the coarsest particles decreases and the proportion of
the finer particles increases continuously until the end of the experiment. The observed changes in the proportions are three
magnitudes larger in Sulvanjoki, where the proportion of coarsest particles decreases strongly between salinities 1 and 3, but
the change slows down to a similar rate as in Laihianjoki above salinity 3 (on the logarithmic scale).

**4.3 Small-volume flocculator experiments**

**4.3.1 Dynamics of major flocculating elements**

In Laihianjoki, flocculator experiments, with discrete salinity treatments, showed that salinity increase induces particle
formation at an almost constant rate throughout the experimental salinity range as indicated by TSM concentration (Fig. 4).
Concentration of organic-bound particulate aluminum ($Al_{org}$) in the organic particulate pool, as determined by the persulfate
digestion, more than doubled from the initial 750 to 1740 µg L$^{-1}$ between salinities 0 to 1.6, above which only moderate
increase was observed. Organic-bound particulate iron ($Fe_{org}$) concentration increased from 585 to 793 µg L$^{-1}$ until salinity
0.8, above which there was no significant increase. POC concentration increased strongly from 5.1 mg L$^{-1}$ at salinity 0 to the
maximum of 8.0 mg L$^{-1}$ at salinity 1.6 and decreased slowly afterwards.

In Sulvanjoki, TSM concentration increased at an almost linear rate throughout the salinity range (Fig. 4). $Al_{org}$ increased at a
similar linear rate from 578 µg L$^{-1}$ at salinity 0 to 4080 µg L$^{-1}$ at salinity 5. $Fe_{org}$ increased from 305 to 560 µg L$^{-1}$ between
salinities 0 and 2.1, above which only a moderate increase was observed. POC increased strongly from 3.7 mg L$^{-1}$ at salinity
0 to the maximum of 8.7 mg L$^{-1}$ at salinity 4.1, above which a small decrease takook place.

The inspection of the proportions of $Al_{org}$, $Fe_{org}$ and POC of TSM provided further insight into the composition of the suspended particle pool. In Laihianjoki, the proportion of $Al_{org}$ increased from 3.2 to 5.2 % of TSM between salinities 0 and 1.6, above which the $Al_{org}$ proportion slowly decreased (Fig. 4). In Sulvanjoki, the $Al_{org}$ proportion showed even stronger increase, doubling from 4.6 to 9.2 % of TSM over the experimental salinity range. In contrast, the $Fe_{org}$ proportion of TSM decreased in both rivers from the initial 2.5 % to 1.7 and 1.3 % at high salinities in Laihianjoki and Sulvanjoki, respectively. Similarly, the POC proportion of TSM in Laihianjoki decreased from 22 % at salinity 0 to 14 % at salinity 4.8. In Sulvanjoki, the POC proportion decreased from 30 % at salinity 0 to 18 % at salinity 5. Clearly, the flocculation rate of Al was higher than that of Fe and organic C, and resulted in the increasing proportion of $Al_{org}$ in the particle pool throughout the experiment.

Apparent particle mean density at 0 salinity was 0.12 mg $L^{-1}$ in Laihianjoki and 0.05 mg $L^{-1}$ in Sulvanjoki (Fig. 5). At salinity 1, Laihianjoki showed a decrease in particle density, but still higher values compared to Sulvanjoki (0.09 mg $L^{-1}$ and 0.04 mg $L^{-1}$, respectively). At salinities above 2, particle density remained relatively stable in both rivers, ranging between 0.03 mg $L^{-1}$ and 0.06 mg $L^{-1}$.

### 4.3.2 Changes in DOM optical properties

UV absorbance ($a_{(CDOM254)}$) and humic-like DOM fluorescence (peak C) decreased strongly in response to increasing salinity in both rivers (Fig. 6). These decreases in aromatic, humic-like DOM compounds were linked with increases in POC, indicating transformation of DOM into particulate form.

Humic-like DOM fluorescence also showed significant inverse relationship with $Al_{org}$ and $Fe_{org}$ concentrations in the suspended particle pool (Fig. 7). In the experiment units without seawater additions, the humic-like fluorescence was the highest, and $Al_{org}$ and $Fe_{org}$ concentrations the lowest. With increasing salinity, humic-like DOM fluorescence decreased, and organic-bound particulate metal concentrations increased.

### 4.3.3 Dynamics of trace metals

Some of the flocculator experiment treatments showed large variability among replicates for trace metal concentrations as determined based on the three replicate analyses (Fig. 8). Therefore, the focus here is on the overall trends and on the metal concentrations in the lowest and in the highest salinity treatments.

Concentrations of organic-bound particulate cobalt ($Co_{org}$) and manganese ($Mn_{org}$) increased in response to salinity increase in both Laihianjoki and Sulvanjoki (Fig. 8). Concentrations of $Co_{org}$ increased from 0.013 µg $L^{-1}$ at salinity 0 to 0.025 µg $L^{-1}$ at salinity 4.8 in Laihianjoki, and from 0.00 µg $L^{-1}$ at salinity 0 to 0.023 µg $L^{-1}$ at salinity 5 in Sulvanjoki. For $Mn_{org}$, there was only a minor concentration increase from 0.56 to 0.64 µg $L^{-1}$ in Laihianjoki, but, in Sulvanjoki, the concentration doubled from 0.19 to 0.37 µg $L^{-1}$.

In contrast, the response of $Cu_{org}$ concentration to the salinity increase was ambiguous (Fig. 8). Concentration of $Cu_{org}$ showed a modest increase from 1.66 µg $L^{-1}$ at salinity 0 to 2.20 µg $L^{-1}$ at salinity 4.8 in Laihianjoki, while it decreased slightly from
2.88 µg $L^{-1}$ at salinity 0 to 2.45 µg $L^{-1}$ at salinity 5 in Sulvanjoki.

### 4.3.4 Flocculation affinity of metals

To estimate the susceptibility of metals to flocculation, the proportion of each metal in the organic particulate fraction was determined compared to its total concentration in the river water (Fig. 9). Using this approach, an increasing proportion value indicated increasing proportion of dissolved fraction transferring into organic particle pool. The studied metals could be
divided into three groups in terms of how they responded to the increasing salinity and pH: (1) Al and Fe with increasing transfer to organic particle pool, i.e. high affinity to flocculation, (2) Cu with high initial affinity to organic particles but no significant response to increasing salinity and pH, and (3) Co and Mn with low affinity to organic particles throughout the salinity and pH gradient.

The flocculation affinity was strongest for Al, whose proportion in the organic particle pool increased from the initial 0–5 to 40–60 % at salinity 5 (Fig. 9). Iron was to a significant degree (25–40 %) associated with organic particles already at the start of the experiment and its particulate proportion increased to 55 % at salinity 5. Both Co and Mn showed low flocculation affinity, as their particulate proportion increased from 0–0.26 to 0.27–0.7 % and from 0–0.27 to 0.10–0.34 %, respectively, throughout the salinity and pH gradient, and in addition showed high variability among replicates. Copper had a relatively
stable particulate proportion of 10–50 %, although the variability among replicates was high.

### 5 Discussion

This study investigated the flocculation of organic matter and metals from dissolved to particulate form in response to mixing particle-free artificial seawater with water from Laihianjoki and Sulvanjoki rivers that are strongly impacted by acidity and metal release from AS soils. A large-volume bucket experiment with continuously measuring *in situ* high-frequency
instruments was run to monitor the development of particle size distribution and organic matter characteristics in the suspended particle pool, and to identify representative salinities for further experimentation. Small-volume flocculator experiments with discrete salinity treatments in a flocculator apparatus were used to investigate metal behavior in flocculation.

At the time of sampling, metal concentrations were extremely high, typical for streams with a high proportion of AS soils, and
the metals were mainly in a dissolved form due to the very low pH (Tables 1 and 2). Sulvanjoki was more acidic and had higher concentrations of Al, Co and Mn, similar concentrations of Cu and lower concentrations of Fe compared to Laihianjoki. On the other hand, higher initial turbidity and lower humic fluorescence (FDOM) in the bucket experiment indicated higher



lithic particle concentration for Laihianjoki (Fig. 2). The differences in metal concentrations and water quality between the rivers were explained by the stronger impact of AS soils on Sulvanjoki due to more intense agriculture in its catchment, and
to the longer flow distance of Laihianjoki (Table 1).

## 5.1 Suspended particle size populations

Development of the suspended particle size distribution in response to seawater additions during the bucket experiments was characterized by two populations with distinct types of behavior: the finer particles, represented by the size class of 11.0 µm median diameter, and the coarsest measured size class of 184.1 µm median diameter (Fig. 3). These size classes agree with
previous laser diffraction-based field studies in the Baltic Sea and elsewhere, which distinguish between particles smaller than 20 µm that are termed *flocculi* and larger particles, known as *flocs* (Mikkelsen et al., 2006; Lee et al., 2012; Asmala et al., 2022). Indeed, Asmala et al. (2022) recently identified similar particle populations of 8 and 184 µm median diameter in the seawater mixing experiments of the Karjaanjoki river (not significantly affected by AS soils) on the Finnish south coast. The 184 µm size class is the coarsest measurable class by the LISST instrument used, which likely underestimates the floc size in
suspension that has been observed to range from tens to thousands of micrometers in field studies (Mikkelsen et al., 2006). For consistency with the previous studies, the terms flocculi and flocs are here adopted to represent the fine and coarsest measured particle populations, respectively.

The bucket experiments showed strong increases in both the volume concentration and proportion of flocs in the suspended
particle pool already at very low salinities in both Laihianjoki and Sulvanjoki (Fig. 3). These increases were accompanied by strong increases in turbidity and strong decreases in humic fluorescence (FDOM) in both rivers (Fig. 2), which strongly indicates intense transformation of DOM into particulate form. Mean particle size in both rivers was well above 80 µm throughout the experiment, which shows that large majority of the newly formed particles were in the floc size range. It is thus concluded that flocs were predominantly formed by the flocculation of DOM. The floc concentration continued to increase
throughout the experimental salinity gradient in Sulvanjoki, whereas the increase was levelled off and turned to decrease at salinity 2 in Laihianjoki. This difference was likely due to the higher concentrations of floc precursors in Sulvanjoki, as shown, for example, by much higher initial FDOM (Fig. 2).

In contrast to the flocs, the concentration of flocculi did not increase until salinity exceeded 1 in Laihianjoki and 2 in Sulvanjoki
(Fig. 3). Instead of a straightforward dependence on salinity, however, the formation of flocculi seems to be controlled also by pH, as it begun when pH exceeded ca. 5.5 in each river. Previous field studies of AS soil-impacted rivers in Finland have shown that Al and Fe are mainly in the dissolved form in pH below ca. 5.5 but are efficiently precipitated as amorphous Al and Fe oxyhydroxides and complexed with organic particles at higher pH when mixing with seawater (Åström and Corin, 2000; Åström et al., 2012; Nystrand et al., 2012). The metal hydroxide composition is in line with previous studies, which
conclude that flocculi are typically single mineral grains or small aggregates of finer mineral grains (Mikkelsen et al. 2006;



Lee et al. 2012). It is thus concluded that flocculi to a significant degree are composed of amorphous Al and Fe oxyhydroxides that start to precipitate at pH above ca. 5.5.

The size population ratio of 184/11 µm highlights the strong initial flocculation of dissolved organic carbon in the salinity range from 0 to 1 in Laihianjoki and particularly in Sulvanjoki (Fig. 3). Above salinity 1, the proportion of organic-dominated flocs decreased and the proportion of oxyhydroxide-dominated flocculi increased continuously until the end of the experiment. Flocculi and small flocs were incorporated in larger flocs with the continuing aggregation (Gregory and O'Melia, 1989; Asmala et al., 2022); however, the increasing proportion of flocculi shows that their precipitation rate exceeded the rate at which they were being incorporated in the flocs (Fig. 3). Indeed, the declining trends in the particle size in both rivers were partially caused 360 by the systematic underestimation of the floc concentration as they aggregated to form particles larger than the LISST measurement range, and the continuous precipitation of flocculi.

## 5.2 Association of Al and Fe with POC

Of the studied elements, the concentrations of Al, Fe and POC were highest in the organic particle pool, and made up 3.2–9.2, 1.3–2.5 and 14–30 % of TSM, respectively, in the flocculator experiments (Fig. 4). These elements are therefore considered 365 to be the main flocculating elements.

The initial concentration of dissolved Al was particularly high in Sulvanjoki, 6.50 mg L$^{-1}$ (Table 2), as was the transformation of Al into the particle form in the experiments. In Sulvanjoki, Al that was bound to organic particles (Al$_{org}$), as determined by the persulfate digestion, increased almost linearly by a factor of 7, from 578 µg L$^{-1}$ at salinity 0 to 4080 µg L$^{-1}$ at salinity 5 370 (Fig. 4). In contrast, in Laihianjoki, dissolved Al concentration was slightly lower, 3.94 mg L$^{-1}$, and Al$_{org}$ increased only by a factor of 2.3 (from 750 to 1740 µg L$^{-1}$) throughout the experiment. The flocculation of Fe was also stronger in Sulvanjoki, where Fe$_{org}$ increased by a factor of 1.8 (from 305 to 560 µg L$^{-1}$) until salinity 2.1, while Fe$_{org}$ increased only by a factor of 1.3 (from 585 to 793 µg L$^{-1}$) until salinity 0.8 in Laihianjoki (Fig. 4). At higher salinities, only modest increases in Fe$_{org}$ were measured for both rivers. The difference in Fe$_{org}$ increase between the rivers was smaller compared to the difference in Al$_{org}$ 375 increase between the rivers, which is likely due to the smaller difference in dissolved Fe concentrations between the rivers (1.4 mg L$^{-1}$ in Sulvanjoki, 1.0 mg L$^{-1}$ in Laihianjoki; Table 2). Indeed, the flocculation rate of Al was much higher than that of Fe, as well as that of DOC, which resulted in the increasing proportion of Al of TSM throughout the experiment, whereas the proportions of Fe and POC decreased (Fig. 4).

POC increased by a factor of 1.6 (from 5.1 to 8.0 mg L$^{-1}$) until salinity 1.6 in Laihianjoki (Fig. 4), resulting in a steep decrease in the apparent mean density of suspended particles from 1.2 mg µL$^{-1}$ to 0.06 mg µL$^{-1}$ in the low salinity range and in a relatively stable and low particle density at higher salinities (Fig. 5). The bucket experiment demonstrated that Laihianjoki had a higher initial proportion of small and dense lithic particles compared to Sulvanjoki, as was shown by the higher initial



turbidity (Fig. 2b), lower initial humic-like FDOM concentration (Fig. 2a) and smaller initial particle mean size (Fig. 2e). It

seems that the strong decrease in the apparent mean density of suspended particles in the Laihianjoki flocculator experiments resulted from the intense formation of low-density organic particles, which exceeded the effect of dense lithic particles on the apparent mean density. A comparable decrease in the particle mean density with increasing salinity was not observed in Sulvanjoki, where the apparent mean density of suspended particles remained roughly at 0.05 mg µL$^{-1}$ despite a relatively stronger increase in POC by a factor of 2.3 (from 3.7 to 8.7 mg L$^{-1}$) until salinity 4.1. The relative stability of the particle

apparent mean density in Sulvanjoki was likely due to the low initial proportion of dense lithic particles.

Organic matter flocculation is known to be selective towards aromatic and humic fractions of organic matter pool (Asmala et al., 2014), which is also apparent in the flocculator experiments. The observed strong decrease in humic-like fluorescence and UV absorbance at 254 nm (Fig. 6) show unambiguously how these fractions are transformed regarding their optical properties

or removed through POC formation. Linkage between decrease in CDOM and increase in POC was more pronounced in Sulvanjoki, suggesting higher susceptibility of organic C from this system for particle formation. The observed very strong coupling between the decrease in humic-like organic matter and the increase in Al$_{org}$ indicates co-precipitation of these dissolved constituents (Fig. 7), instead of e.g. preferential organic matter adsorption to lithic particles already present. Coupling of organic matter and Fe$_{org}$ was weaker, which suggests that the complexation of Fe with organic matter is less important in

overall floc formation. Together with the rapid increase in the floc and flocculi ratio (Fig. 3) and the decrease in particle density (Fig. 5), it is evident that organic-dominated flocs are the preferred pathway for especially Al, and to some extent for Fe particle formation.

The higher affinity of Al than Fe to organic matter during flocculation was also reflected in the composition of the organic

particle pool. The proportions of Al and Fe in the organic particle pool increased by 35 and 15 %, respectively, in Laihianjoki, and as much as 60 and 30 %, respectively, in Sulvanjoki over the salinity range from 0 to 2 (Fig. 9). Such strong transfer to particulate form indicates efficient capture of dissolved Al and Fe in organic-dominated flocs, whose formation is the principal flocculation process in the lowest salinity range (Section 5.1). The strong organic flocculation affinities of Al and Fe are in line with previous flocculation studies of boreal rivers that typically have high DOC and Fe concentrations (Pettersson et al.,

1997; Yu et al., 2015; Jilbert et al., 2018; Khoo et al., 2022), as well as AS soil-impacted rivers that also have high Al concentrations (Nordmyr et al., 2008a, 2008b; Nystrand and Österholm, 2013; Nystrand et al., 2016).

In addition to the floc formation, Fe is to a lesser extent precipitated as oxyhydroxides, particularly ferrihydrite, during the mixing with seawater of Finnish AS soil impacted rivers, while hydroxysulfate minerals are largely absent (Yu et al., 2015).

Accordingly, the increasing concentrations of flocculi, beginning when pH exceeds ca. 5.5 at salinity 1 in Laihianjoki and 2 in Sulvanjoki, indicate that the precipitation of Al and Fe hydroxides, as well as the potential Al and Fe adsorption on the oxyhydroxide surfaces, also contribute to the particulate Al and Fe pool (Bigham and Nordstom, 2000; Åström and Corin,





2000). However, the low relative concentration of flocculi implies that Al and Fe hydroxides are minor components of the particulate Al and Fe pool.

### 5.3 Flocculation behavior of trace metals Co, Mn and Cu

The proportions of Co and Mn associated with organic particles were <1 and <0.4 %, respectively, and increased only slightly with salinity and pH in both Laihianjoki and Sulvanjoki (Fig. 9). These observations agree with previous geochemical modeling and field studies in Finnish AS soil impacted rivers, which have shown that Co and Mn are weakly associated with organic complexes and are relatively persistent in solution during the mixing with seawater in estuaries (Åström and Corin, 2000; Nystrand and Österholm, 2013; Nystrand et al., 2016). The observations are further supported by previous laboratory studies that have shown Mn and Co binding to humic substances to be relatively minor (Sholkovitz, 1978; Hamilton-Taylor et al., 2002). However, the proportions of Mn and in particular Co in the particle pool did increase slightly towards the end of the experiments. Both Mn hydroxide precipitation and Co complexation with organic substances are comparably slow processes and require a pH >6 (Pontér et al., 1992; Moffet and Ho, 1996; Turner and Mawji, 2005), which was reached only at the end of the experiments (Fig. 3). It is, therefore, probable that longer experiment time or stronger increase in salinity would have resulted in more efficient transfer of Mn and Co to the particle pool. It is worth noting, however, that salinity in the recipient Gulf of Bothnia is lower than the maximum salinity 6 of the experiment.

Previous studies in the nearby Vöyrinjoki estuary have concluded that Co co-precipitates with Mn oxyhydroxides (Nordmyr et al., 2008a, 2008b; Nystrand et al., 2016), whereas studies elsewhere have concluded Co flocculation to be driven by binding to organic material and decoupled from Mn oxyhydroxide precipitation (Moffet and Ho, 1996). Indeed, the transfer of Co to the particle pool was roughly twice of that of Mn in both studied rivers, which suggests that Co flocculation is at least partly decoupled from Mn oxyhydroxide precipitation.

The proportion of Cu in the organic particle pool varied between 10 and 50 % during the experiment in both rivers (Fig. 9), which is comparable to other AS soil impacted rivers in Finland where up to 60 % Cu is found in strong association with humic particles (Pettersson et al., 1997; Nystrand & Österholm, 2013; Nystrand et al., 2016). It is notable that no additional net transfer of Cu to the organic particle pool was observed as a response to the increasing salinity and pH in either river. Also previous laboratory experiments have shown that the binding of Cu to organic ligands, particularly humic substances, may be unaffected by increases in salinity in estuaries (Hamilton-Taylor et al., 2002). Interestingly, up to 56 % of Cu occurs complexed possibly with organic matter in the Gulf of Bothnia (Bordin et al., 1988). It appears that a substantial proportion of Cu is bound to organic particles in the studied acidic and humic-rich rivers, and this proportion is not significantly changed during mixing with seawater.



**5.4 Implications for seaward metal transport**

Previous field and geochemical modelling studies in the west coast of Finland have concluded that when acidic metal-rich river waters from the boreal AS soil landscape are discharged to estuaries with higher pH and salinity, Al and Fe to a large extent form complexes with flocculating organic matter and precipitate as Al and Fe oxyhydroxide minerals that are deposited in sediments close to the river mouths (Nordmyr et al., 2008a, b; Nystrand et al., 2016). Flocculator experiments carried out here showed relatively stronger complexation with the organic-dominated flocs for Al (Section 5.2), which may explain the

higher relative Al contents in seafloor sediments in the inner archipelago compared to outer archipelago (Fig. 1c). In particular, Al was shown to be enriched above its local background level in the innermost sediment core MGGN-2016-8 (Virtasalo et al., 2020), confirming the mechanism of early flocculation and deposition at short distance from these rivers of Al.

Mn and Co were transferred to the particulate form only in small amounts in the experimental seawater treatments. Previous

studies have shown that Mn hydroxide precipitation and Co complexation with organic substances are slow processes and take place at a comparably high pH above 6 (Pontér et al., 1992; Moffet and Ho, 1996; Turner and Mawji, 2005), which suggest that the previously proposed mechanism of Mn precipitation and deposition as Mn oxyhydroxides farther out at the sea may be the controlling mechanism of seafloor Mn distribution (Nordmyr et al., 2008a, b; Nystrand et al., 2016). Co transfer to particulate form was twice of that of Mn, which highlights Co complexation with flocs in addition to Mn oxyhydroxides. This

interpretation is supported by the seaward enrichment patterns of Co and Mn in the front of Laihianjoki and Sulvanjoki, where both Co and Mn are strongly enriched up to 12 km distance, while a lower Co enrichment continues to at least 26 km distance (Virtasalo et al., 2020).

Up to 50 % of total river-borne Cu was bound to humic substances in flocs in Laihianjoki and Sulvanjoki (Fig. 9), which is

similar to other boreal AS soil-impacted rivers draining to the Gulf of Bothnia (Pettersson et al., 1997; Nystrand & Österholm, 2013; Nystrand et al., 2016). This proportion was not significantly changed due to mixing with seawater in the flocculator experiment. Previous studies in the nearby Vöyrinjoki estuary have documented that Cu to a large part is deposited in association with POC close to the river mouth (Nordmyr et al., 2008a, b; Nystrand et al., 2016). However, minor Cu enrichment in sediments at least 26 km seaward from the Laihianjoki and Sulvanjoki river mouths require that some Cu is transported and

deposited farther out at the sea (Virtasalo et al., 2020). Previous studies have concluded Cu binding to organic ligands to be a significant mechanism for Cu distribution in the Gulf of Bothnia and elsewhere (e.g., van den Berg et al., 1987; Bordin et al., 1988; Muller and Batchelli, 2013).

A previous geochemical study of seafloor sediments in the front of Laihianjoki and Sulvanjoki found that AS soil-derived

metals such as Cu and Co are strongly positively correlated with organic matter, particularly in the 2–6 µm sediment grain size range (Virtasalo et al., 2020). Also Al showed a weak correlation with this grain size range. Organic-dominated flocs that were

formed during the bucket experiment here were generally larger than 80 µm, and the flocculi were predominantly composed of Al and Fe oxyhydroxides (Section 5.1). It seems that the floc size in suspension does not correspond to the size of metal-rich organic grains in the sediments, which is probably the consequence of the post-depositional break-up of flocs to their smaller constituent particles (Mikkelsen et al., 2006; Lee et al., 2012). The large and soft flocs are easily broken after deposition by organic matter degradation and sediment compaction with burial, and by sediment mixing by benthic organisms (e.g. Rhoads and Boyer, 1982).

Climate change is predicted to result in increasing temperatures and evapotranspiration during summer in Finland (Olsson et al., 2015). This effect is likely to enhance the drying and oxidation of sulfides in AS soils (Österholm and Åström, 2008; Job et al., 2020) and increase the acidic runoff and metal loading of AS soils to the recipient water bodies (Saarinen et al., 2010; Nystrand et al., 2016). The results of the flocculator experiments suggest that climate change will increase the transport of Co, Cu and Mn, and to a smaller spatial extent of Al, to the Gulf of Bothnia, while the Fe transport will be less affected.

## 6 Conclusions

This study shows that mixing of acidic and dissolved metal-rich waters from boreal AS soil-impacted rivers with seawater induces strong flocculation of dissolved organic matter to the particulate form already at salinity 0–2. Al and Fe were strongly transferred to the newly formed organic-dominated flocs that were generally larger than 80 µm. Al transfer to the flocs was stronger than that of Fe, which is also reflected in the previously documented relatively higher Al contents in the seafloor sediments of inner parts of the recipient archipelago of the studied rivers. The flocs in the suspended particle pool were complemented by a smaller population of Al and Fe oxyhydroxide-dominated flocculi (median size 11 µm) after pH exceeded ca. 5.5. Cobalt and Mn transfer to the particle pool was weak, although some transfer to Mn oxyhydroxides as well as Co association with flocs took place. Up to 50 % Cu was found to be bound to humic substances in flocs in the AS soil-impacted rivers and this proportion did not significantly change during mixing with seawater.

This study highlights that, in addition to salinity, pH should be considered in estuarine flocculation and coastal filter studies, particularly when AS soils are present in the catchment area.

*Author contributions.* JV conceptualized the study. EA planned and led the laboratory experiments. PÖ identified the sampling locations. JV and EA conducted the formal analysis and wrote the original draft. PÖ contributed to the reviewing and editing of the manuscript.

*Competing interests.* The authors declare that they have no conflict of interest.



*Acknowledgements.* This study has utilized research infrastructure facilities provided by the Finnish Marine Research Infrastructure (FINMARI) network. Help from the laboratory personnel at the Tvärminne station is gratefully acknowledged.

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

**Figures**

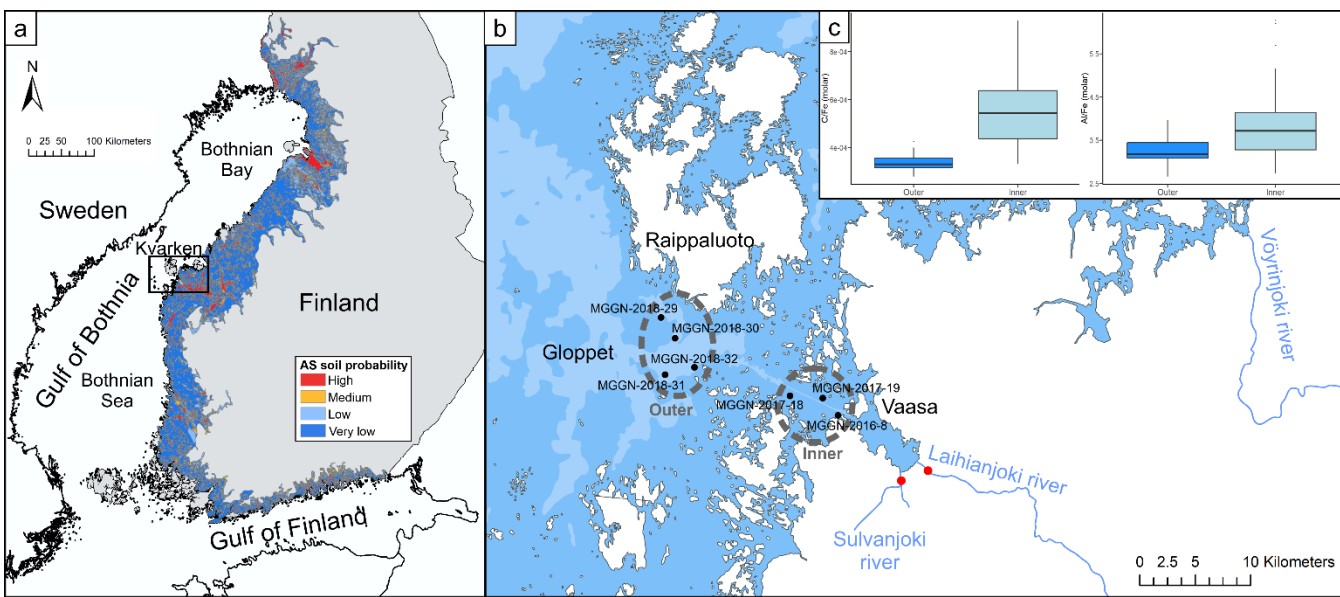

**Figure 1. Maps of the Baltic Sea and the study area, with a boxplot of molar Al, Fe and C ratios in the inner and outer archipelago.**
**(a) Map of the Baltic Sea and the probability of acid sulfate soil occurrence in Finland. Black square indicates the location of the**
**study area on the west coast of Finland. (b) Nautical chart of the study area in the Kvarken archipelago. Red dots indicate the water**
**sampling locations of this study. Black dots indicate the coring sites of sediment samples from Virtasalo et al. (2020) that were used**
**for the molar element ratio calculations in the inner and outer archipelago. (c) Molar C/Fe and Al/Fe ratios in the inner and outer**
**archipelago. Sources: acid sulfate soil probability map – Geological Survey of Finland, 2018; nautical chart – Finnish Transport**
**Agency, 2017.**







**Figure 2. Humic-like fluorescence (FDOM), turbidity, pH, particle total volume concentration and particle mean size in suspension**
**during the bucket experiment.**

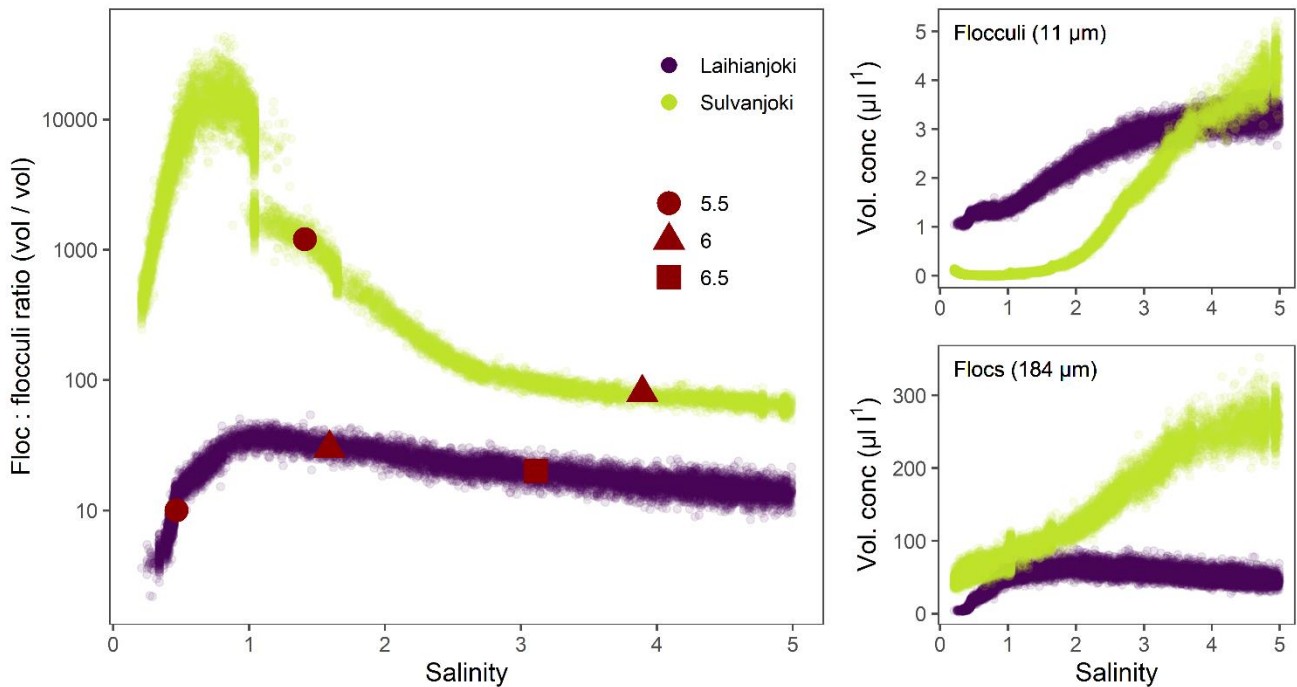

**Figure 3. Volume concentrations of the 11.0 µm (flocculi) and 184.1 µm (flocs) median size populations in the bucket mixing experiment. Relevant pH values are indicated in red symbols in the left panel.**



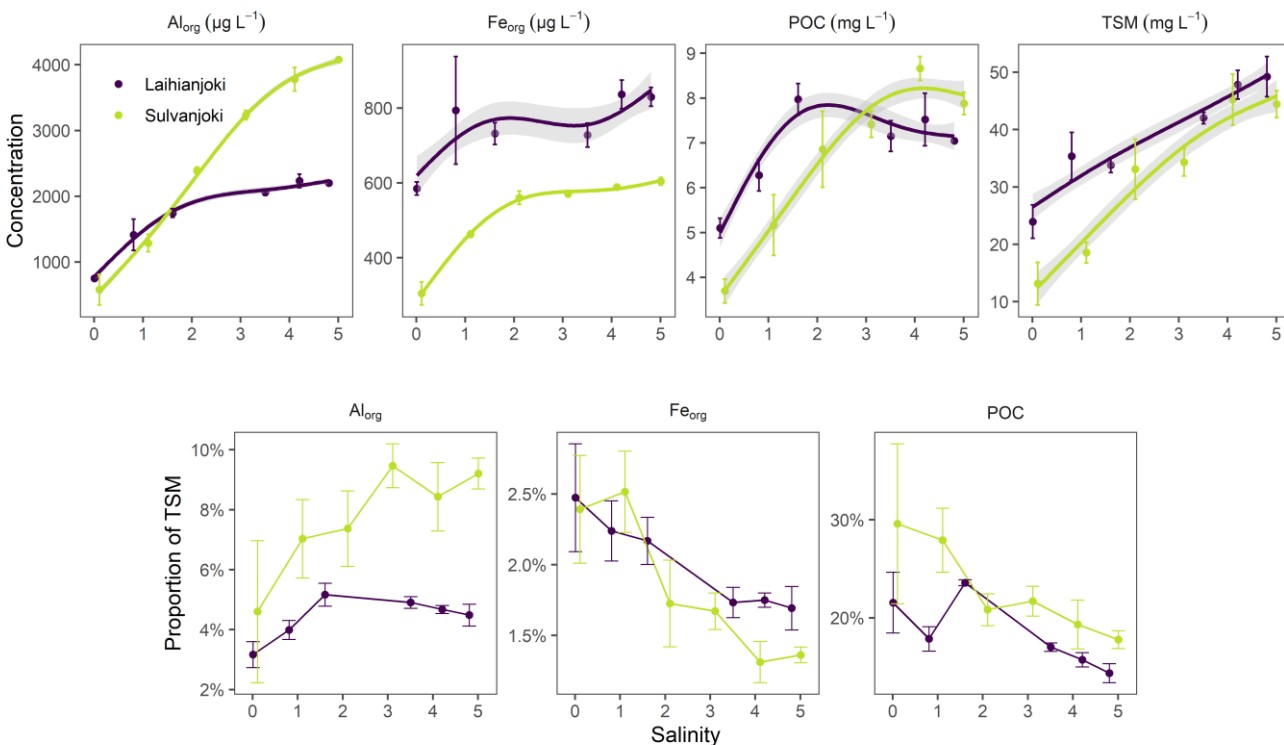

**Figure 4. (top row) Concentration of Al$_{org}$, Fe$_{org}$ (µg L$^{-1}$) and POC (mg L$^{-1}$) in the organic particle pool and TSM in the bucket experiment. (bottom row) Proportions of Al$_{org}$, Fe$_{org}$ and POC of TSM in the bucket experiment. Points denote mean values of three replicates and errorbars 1 standard deviation.**





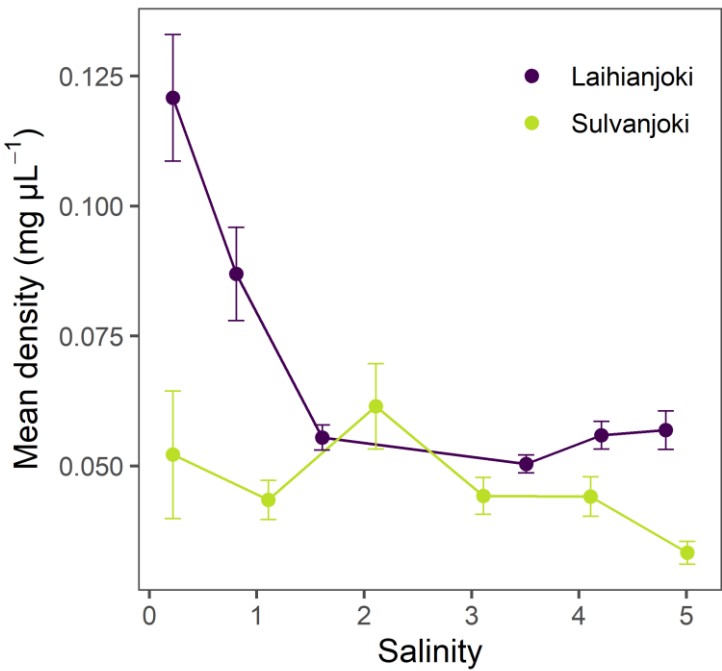

**Figure 5. Apparent mean density of suspended particles in the flocculator experiments along the artificial salinity gradient. Points denote mean values of three replicates and errorbars 1 standard deviation.**

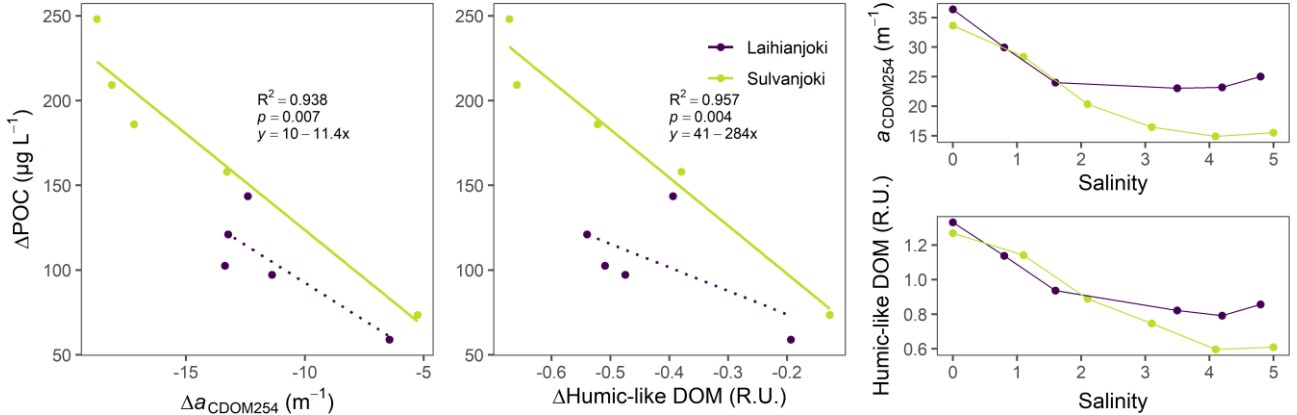

**Figure 6. Relationship between changes in POC concentration and a) CDOM absorption coefficient at 254 nm ($a_{CDOM254}$), and b) humic-like fluorescence (peak C; ex350/em420-480 nm) in the flocculator experiments. Significant (p < 0.05) linear relationship is indicated with a solid line and non-significant with a dotted line. Changes of c) $a_{CDOM254}$ and d) humic-like fluorescence along the salinity gradient of the mixing experiments is also given.**



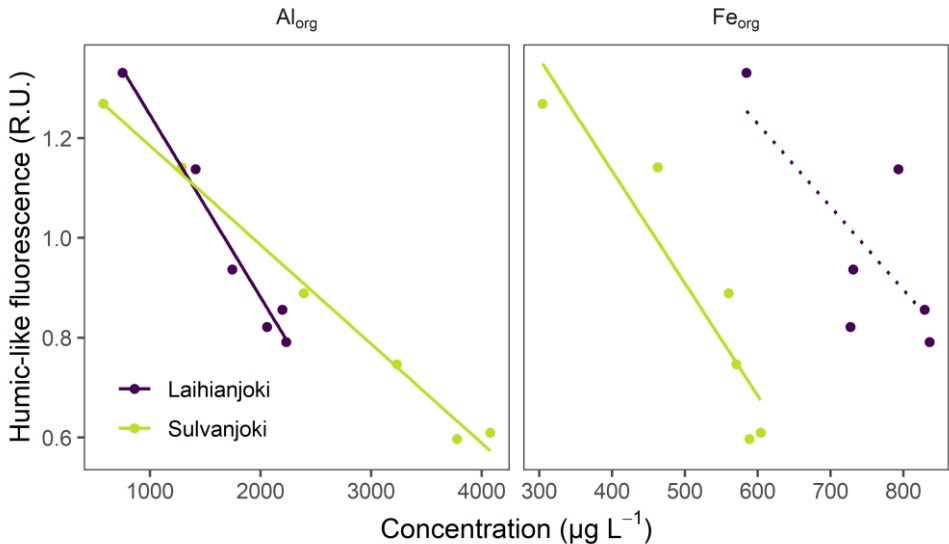

**Figure 7. Relationship between the concentrations of Al_org and Fe_org (mg L$^{-1}$), and humic-like fluorescence (peak C; ex350/em420-480 nm) in the flocculator experiments. Significant (p < 0.05) linear relationship is indicated with a solid line and non-significant with a dotted line.**

790

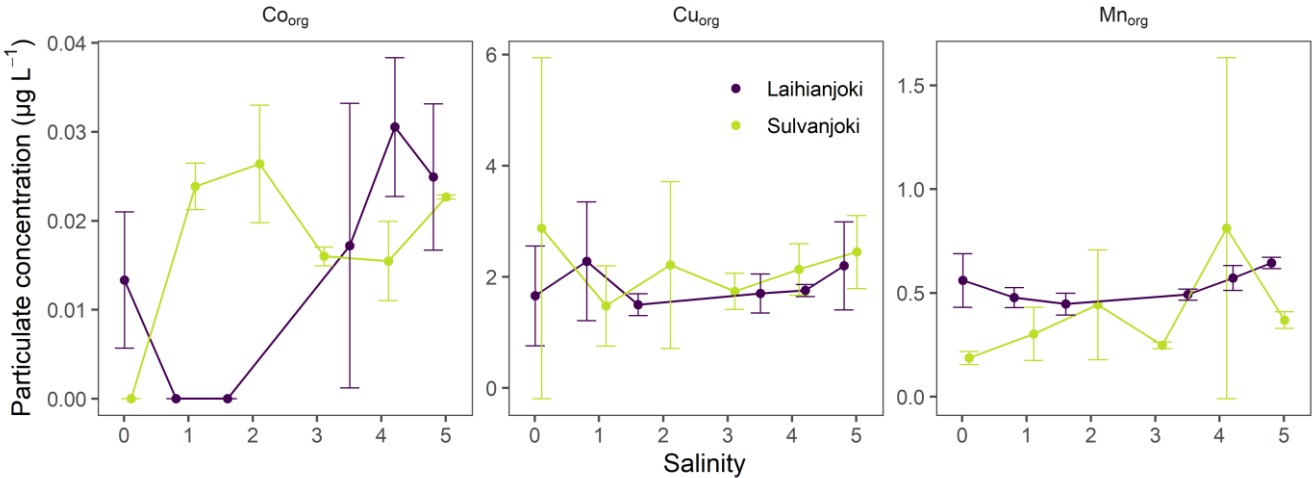

**Figure 8. Concentrations of Co_org, Cu_org and Mn_org (µg L$^{-1}$) in the flocculator experiments. Points denote mean values of three replicates and errorbars 1 standard deviation.**

795



**Figure 9. Flocculation affinity of Al, Fe, Cu, Co and Mn, as indicated by change in the proportion of particulate fraction of total concentration along the experimental seawater gradient. Solid line indicates significant (log-)linear trend (p < 0.05) and dotted line non-significant. Points denote mean values of three replicates and errorbars 1 standard deviation.**

800





**Tables**

Table 1. Properties of river catchments and mean annual values of discharge (Q), total Al concentration, total Fe concentration and molar Al:Fe ratio (Korhonen and Haavalammi, 2012). "Agriculture" includes pasture and cropland, "open" consists of areas with no vegetation and "water" includes lakes, streams, etc. Details about sampling location and water chemistry during sampling are also given.

|  | Laihianjoki | Sulvanjoki |
|---|---|---|
| General information | | |
| Catchment area (km$^2$) | 506 | 144 |
| Length (km) | 60 | 33 |
| Land use (%) | | |
| Urban | 6 | 6 |
| Agriculture | 27 | 36 |
| Forest and peatland | 61 | 56 |
| Open | 5 | 2 |
| Water | 1 | 0.1 |
| Mean Q (m$^3$ s$^{-1}$) | 3.1 | 1.3 |
| Mean pH | 4.4 | 4.0 |
| Mean total Al (mg L$^{-1}$) | 2.81±1.35 | 5.77±4.87 |
| Mean total Fe (mg L$^{-1}$) | 2.57±1.26 | 2.26±2.06 |
| Mean Al:Fe ratio (mol mol$^{-1}$) | 0.59±0.26 | 3.42±6.79 |
| Sampling conditions | | |
| Date | 13 April 2021 | 13 April 2021 |
| Latitude (°N; WGS84) | 63.043 | 63.034 |
| Longitude (°E; WGS84) | 21.705 | 21.660 |
| pH | 4.75 | 4.56 |
| Water temperature (°C) | 3.2 | 3.1 |
| Conductivity (µS cm$^{-1}$) | 275 | 401 |
| Salinity | 0.1 | 0.2 |

Table 2. Metal concentrations in the Laihianjoki and Sulvanjoki river water samples and artificial seawater.

|  | n | Al (mg L$^{-1}$) | Fe (mg L$^{-1}$) | Co (µg L$^{-1}$) | Cu (µg L$^{-1}$) | Mn (µg L$^{-1}$) |
|---|---|---|---|---|---|---|
| Laihianjoki | 3 | 3.94±0.04 | 1.40±0.03 | 18.9±0.5 | 13.3±0.4 | 658±18 |
| Sulvanjoki | 3 | 6.50±0.13 | 1.01±0.04 | 30.5±0.2 | 13.7±0.2 | 797±4 |
| Artificial seawater | 3 | <0.2 | 0.07±0.01 | 0.6±0.1 | 11.1±0.8 | 26±2 |



## Appendix A

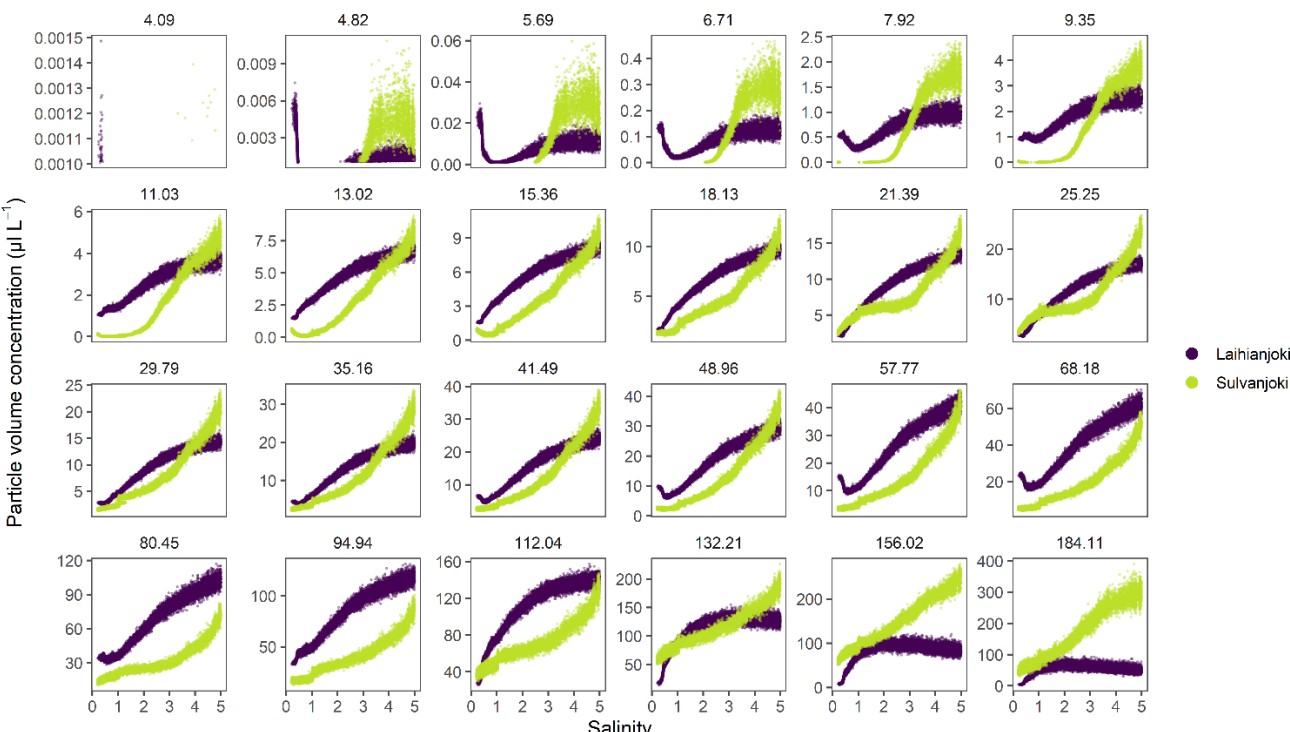

840    **Figure A1. Volume concentrations of LISST-100X particle size classes (bins) in the bucket experiment. The size classes are labeled by their median size (μm). The finest eight (1.18–3.76 μm) of altogether 32 size classes are excluded because they displayed very low concentrations and were not considered further.**