# Peer review of "Estuarine flocculation dynamics of organic carbon and metals from boreal acid sulphate soils"

_Biogeosciences, 2023_

## Author Response (AR1)

Dear Editor,

Thank you for considering our manuscript "Estuarine flocculation dynamics of organic carbon and metals from boreal acid sulphate soils" for publication in *Biogeosciences*.

We have now revised the manuscript according to the comments by two anonymous referees. In general, we found the comments justified and helpful with respect to improving the manuscript.

Below we answer all review comments point by point. Referee comments are in **black**, and our responses in blue.

A track-changes version of the revised manuscript is appended at the end of this document.

Line numbers given in brackets **[]** in our responses refer to the line numbers in the marked-up manuscript.
* * *
**Referee #1:**

**This work investigated the estuarine flocculation dynamics of organic carbon and metals from boreal acid sulphate soils. This process is critical for the cycling of OM and metals in aquatic environments. I have several major comments for this manuscript and they are listed below.**

We thank Referee for the useful comments that have helped improve the manuscript.

**Specific comments:**

**Introduction: previous studies on estuarine flocculation dynamics of organic carbon and metals should be reviewed.**

We would like to point out that the third paragraph of Introduction already summarizes the findings of previous studies on the estuarine flocculation dynamics of organic carbon and metals in the context of acid sulphate soils **[Lines 61-73]**. Perhaps Referee means that we should review the estuarine flocculation dynamics of organic carbon and metals in general? For that case, we have added a sentence saying that differing physical chemistries of trace metals lead to a variety of geochemical behaviours in the estuarine flocculation and included relevant case studies as references **[Lines 43-45]**. We find that a detailed review of the estuarine flocculation of organic carbon and metals in general would be unnecessary or even confusing because flocculation in the context of acid sulphate soils is a special case where, in addition to the usual salinity increase, the strong increase in pH influences the geochemical behaviour of metals.

**Line 140: what is the material of the bucket used? Will it adsorb the metals?**

We now mention that the bucket was made of polypropylene. **[Line 143]**

Metal flocculation was studied in the flocculator experiments that used acid-washed glass beakers, so the bucket material is not relevant for the metals.

**Line 175: why were different instruments adopted to measure these metals?**

We added a mention in the text that different instruments were used for the selected metals for improved analytical performance. **[Line 183, 204]**

**Section 4.1: this section is too short and it need to be merged into the other sections.**

Referee is correct that the section is only one sentence. However, for the sake of clarity, we find it is good that the initial compositions of the river waters are considered separately from the experiment results and, therefore, prefer leaving this section as it is. Because this is a structural matter, we expect Editor to provide us with guidance if necessary. **[Lines 219-220]**

**Line 205: 'The initial DOM concentration, as indicated by humic-like fluorescence', why not directly measure the DOC?**

High-frequency optical sensors for determining DOC concentration are not available, whereas sensors for determining DOM fluorescence are commercially available and have been widely used as a proxy for DOC concentration for the past decades. We have added a reference to a recent review article about this topic. **[Line 356]**

**Line 310: This paragraph could be deleted as it repeated with previous statements.**

Referee is correct that the first paragraph of Discussion briefly restates the study aim and methodological approach. The purpose of the paragraph is to help the busy reader who may wish to jump to Discussion section and, therefore, we would like to keep it as it does not take up much space. Because this is a structural matter, we expect Editor to provide us with guidance if necessary. **[Lines 326-331]**

**Line 490: 'The results of the flocculator experiments suggest that climate change will increase the transport of Co, Cu and Mn, and to a smaller spatial extent of Al, to the Gulf of Bothnia, while the Fe transport will be less affected.', how can this conclusion be obtained?**

We have clarified our reasoning in that sentence, and it now reads "The flocculator experiments showed that Co, Cu and Mn largely remain dissolved during estuarine mixing, which will favor the seaward transport of their riverine concentrations that are expected to increase due to climate change. Aluminum was more strongly transferred to organic-dominated flocs than Fe, and the flocs provide a mechanism for the transport of increasing riverine Al concentrations to the innermost archipelago. Iron is not enriched in the Gulf of Bothnia by loading from the boreal AS soils and this condition is unlikely to change due to climate change." **[Lines 515-521]**

This ends the comments by Referee #1

Referee #2:

**In this study, Virtasao et al. use laboratory experiments to simulate the flocculation of several metals during the drainage of boreal acid sulfate soils into seawater. They used a combination of bucket experiment with continuous, in situ monitoring, and smaller-scale flocculator with more detailed chemical analysis. The results have clear implications for the export of dissolved metals from terrestrial to marine environments.**

**In general, I found that the results were presented very clearly and the conclusions were well-supported.**

We thank referee for the encouraging and useful comments that have helped improve the manuscript.

**My only potential major concern is that the approach to replication needs to be clarified. Line 162 indicates that at the end of the flocculator experiments, suspended material was collected on triplicate filters. As written, this sounds like psuedoreplication (repeated sampling of a single, unreplicated experiment).**

We have now clarified throughout the text that by replicates we mean analysis results of triplicate filter retentate samples from the flocculator experiments. **[Lines 198, 209, 263, 297, 322, 323, 826, 831, 879, 886]**

**It doesn't appear that the bucket experiments were replicated, which is probably okay given the nature and purpose of these experiments, but that should be clarified in the text.**

It is mentioned twice in the text that only a single bucket experiment was carried out: Methods section **[Line 137]** and at the beginning of Discussion **[Line 328]**

**Other comments:**

**Lines 87-91: comparison between the two rivers is an important aspect of the study. I suggest adding an object and/or hypothesis related to this comparison.**

We added a third hypothesis "differences in metal concentrations and water quality between the studied rivers affect the flocculation behaviour of DOM and metals" in Introduction. **[Line 93-94]**

**Lines 134-151: a few details are missing in the setup of the bucket experiment: what was the initial volume of artificial seawater, what was the rate of river water addition, and what was the total duration of the experiment?**

We have added the missing details about the bucket experiment: the initial volume of river water, the rate of seawater addition, and the total duration of the experiment. **[Lines 142, 144-145]**

**Line 335: "already at very low salinities" this sentence is not clear.**

We now state the relevant salinity range 0-2. **[Line 354]**

**Lines 391-402: I don't think the data demonstrate selective removal of aromatic or humic-like DOM. To demonstrate selectivity you would need to show that these fractions are removed at a higher rate than bulk DOM. Figure 6 shows that aromatics and humics are removed, but not necessarily selectively.**

We have added two new figures (Fig. A2 and A3) in Appendix that show selective removal of humic-like DOM through POC formation. This observation is also briefly mentioned in the text **[Lines 415-416].**

[Figure]

**Figure A2.** Original CDOM fluorescence excitation-emission matrix (left) and changes in fluorescence intensity due to salt addition (right). Negative change in fluorescence intensity indicates removal of fluorophores, in this case substances with humic-like fluorescence. Humic-like fluorescence peaks (A and C) are marked with red rectangles, and protein like peak (T) with a black rectangle. Peaks defined following Coble (1996). In essence, the largest decreases in CDOM fluorescence are observed in the humic-like substances, whereas removal of protein-like substances is negligible.

[Figure]

**Figure A3**. Changes of $a_{CDOM254}$ (top) and $a_{CDOM440}$ (bottom) along the salinity gradient in the flocculator experiments. To summarize, UV-absorbing, aromatic molecules have more pronounced and coherent response to increasing salinity, compared to VIS-absorbing, non-aromatic (aliphatic) organic molecules.

**Line 430: "longer experiment time…" this is why it is important to clarify the length of the experiment in the methods section!**

We agree. The experiment duration is now added in Methods. **[Line 145]**

**Line 436: what is meant by the local background level here?**

We have clarified the sentence by comparing the current Al concentration to typical levels in sediments deposited before the intensive artificial drainage of the AS soil landscape started in the 1960s. **[Lines 477-478]**

**Figures: panel letters are provided for Figure 1 but not the rest of figures. In general, it would be helpful to include these in all figures and refer to them in the text.**

We have added panel letters to all figures and use them when referring to the figures in the text.

**Figure 4: the text indicates these results are from the flocculator experiments but the caption indicates bucket experiment.**

We thank Referee for pointing this out. Indeed, results in the figure are from the flocculator experiments. The mistake has been corrected. **[Line 856]**

**Figure 9: several of the data points and error bars seem to indicate a negative percentage of the metals are found in the particulates. How is this possible?**

We had a blank subtraction error regarding Mn, resulting in values slightly below zero for two samples with very low concentrations. This has now been corrected. In addition, we have updated Figure 9 by showing each measurement as points (n=3) instead of means and error bars.

This ends the comments by Referee #2, and our responses to the peer-review comments.

Kind regards on behalf of all co-authors,

Joonas Virtasalo, Geological Survey of Finland (GTK)